# Adverse Childhood Experiences, Social Isolation, Job Strain, and Cardiovascular Disease Mortality in U.S. Older Employees

**DOI:** 10.3390/medicina59071304

**Published:** 2023-07-14

**Authors:** Timothy A. Matthews, Jian Li

**Affiliations:** 1Department of Environmental Health Sciences, Fielding School of Public Health, University of California Los Angeles, 650 Charles E. Young Drive South, Los Angeles, CA 90095, USA; tmatthews@ucla.edu; 2School of Nursing, University of California Los Angeles, 650 Charles E. Young Drive South, Los Angeles, CA 90095, USA

**Keywords:** adverse childhood experiences, job strain, social isolation, cardiovascular disease mortality, older employees

## Abstract

Stress is a key driver of cardiovascular disease (CVD), yet the contribution of psychosocial stressors to the development of CVD has not been systematically examined in United States (U.S.) populations. The objective of this study was to assess prospective associations of adverse childhood experiences (ACEs), social isolation, and job strain with CVD mortality. Data were from the large, nationally representative, population-based Health and Retirement Study (HRS). ACEs, social isolation and job strain were assessed using validated survey instruments at baseline between 2006–2008, and death information was followed up through 2018. Cox proportional hazards regression models were used to examine prospective associations of ACEs, social isolation, and job strain with CVD mortality among 4046 older employees free from CVD at baseline. During 42,149 person-years of follow-up time, 59 death cases of CVD were reported. After adjustment for covariates, ACEs and job strain were significantly associated with increased risk of CVD mortality (aHR and 95% CI = 3.67 [1.59, 8.48] and 2.24 [1.21, 4.11], respectively), whereas social isolation demonstrated an inflated but nonsignificant association (aHR and 95% CI = 1.62 [0.72, 3.66]). These findings highlight the role of psychosocial exposures as novel and clinically relevant risk factors for CVD.

## 1. Introduction

Cardiovascular disease (CVD), including heart disease and stroke, is the leading cause of death worldwide and in the United States (U.S.) [1]. The prevalence of CVD in U.S. adults is 49%, and the American Heart Association’s (AHA) 2023 update on Heart Disease and Stroke Statistics reported 928,741 CVD deaths in the US in 2020 [1]. Furthermore, the World Health Organization’s (WHO) projections estimate that CVD will account for over 22.2 million annual deaths by 2030 [2]. The economic burden of CVD in the U.S. is estimated at over $350 billion annually, with a projected increase to over $950 billion by 2035 [1,3]. CVD broadly impacts demographic groups, affecting men and women of all ages and racial backgrounds [4]. While CVD was previously considered a disease of aging populations, the incidence of CVD in younger persons and working populations is rapidly increasing [5]. Additionally, although CVD mortality in the U.S. has declined over the course of the past two decades, the rate of decline has stagnated, and CVD continues to exert a heavy mortality burden, with recent data showing an increase again in the later 2010s to 2020 [1,6,7,8].

Traditional risk factors such as unhealthy diets, smoking, and physical inactivity are unable to fully account for the burden of CVD risk, and recent evidence has identified an urgent need to evaluate the contribution of novel environmental, behavioral, and occupational risk factors to CVD [9,10,11,12]. Accordingly, the U.S. National Heart, Lung, and Blood Institute’s (NHLBI) Strategic Vision for Research has called for the targeted epidemiological investigation of novel risk factors for CVD [13]. Such risk factors can be broadly classified as stressors, and a rapidly emerging body of evidence has posited a central role of stress in the etiology and pathophysiology of CVD [14,15]. 

Research efforts have identified a continuum of novel stressors that contribute to CVD across all stages of the life-course, from childhood through adulthood and old age. Notably, psychosocial exposures are a major source of stress and increased CVD risk across the life-course, the effects of which can be dissected according to their associated life stage [16,17,18]. Adverse childhood experiences (ACEs), defined as a constellation of early life adversities such as childhood abuse, neglect, or household dysfunction, have emerged as a severe and causal risk factor for CVD [19,20,21], while in adulthood, social isolation [16,22] and job strain have demonstrated extensive and causal associations with CVD [17,23,24,25,26]. This collective set of stressors represents a holistic spectrum of work and nonwork related psychosocial stressors evidenced as CVD risk factors, across the life course. While studies have demonstrated the separate associations of such psychosocial exposures with CVD outcomes, there is an extreme lack of evidence covering these stressors in one single study. Importantly, the systematic and integrated investigation of these novel stressors and their aggregate contributions to CVD at different life stages demands a novel conceptual framework. Most recently, the life-course model has emerged as a theoretical framework for effectively characterizing the differential effects of exposures across life stages [27,28,29]. The life-course model has benefitted additionally from integration with the cumulative advantage/disadvantage model, which was developed to assess the issue of multiple overlapping exposures with potential additive or synergistic effects [30,31]. 

While some pioneering research efforts have identified interactions of job strain with other psychosocial factors in the context of mental health outcomes [32,33,34], no study has examined the comprehensive associations of ACEs, social isolation, and job strain with risk of CVD, presenting a research gap. A single study in 2007 assessed the contribution of early-life risk factors (in this case, parental socioeconomic status) to the association of job strain with atherosclerosis, stating that “even the best-designed prospective studies have failed to take into account the cumulative effects of early life factors”, but found no evidence for such effects [35]. Therefore, this project seeks to address these knowledge gaps via the systematic investigation of work and nonwork related psychosocial exposures across the life course. 

The objective of this project is to conduct a systematic epidemiological investigation of ACEs, social isolation, job strain, and their association with CVD mortality in a large, nationally representative, population-based sample of U.S. older employees. This analytic project based on the Health and Retirement Study (HRS) examines the contribution of psychosocial stressors at different life stages (childhood and adulthood) and in different life domains (working life and non-working life) to death from CVD. There is a critical knowledge gap regarding the interaction of work and nonwork related psychosocial stressors in cardiometabolic disease processes. While many studies have evaluated the separate contributions of either work or nonwork related exposures, there is a paucity of empirical evidence assessing their interrelationships, and hence the National Occupational Research Agenda (NORA) has highlighted a need for research on the “relationship of occupational risk factors with known non-occupational risk factors for CVD” [36]. This project implements a comprehensive exposure assessment model incorporating the work and nonwork related psychosocial factors of ACEs, social isolation, and job strain, from childhood to adulthood and old age, spanning the entirety of the life-course. Our central hypothesis is that ACEs, social isolation, and job strain lead to increased risk of CVD mortality.

## 2. Materials and Methods

### 2.1. Sample Population

The data for this research project were derived from the HRS cohort—a large, nationally representative, population-based sample of U.S. adults aged 50+. Analyses were focused on individuals with employment experience, while participants with self-reported physician-diagnosed CVD (including heart disease and stroke) at baseline were excluded to produce accurate estimates of CVD mortality during follow-up and to eliminate the impact of pre-existing CVD on future CVD mortality. Data from the HRS study were sourced from public-use datasets [37,38,39,40,41,42]. Follow-up time began in 2006–2008, the baseline time-point for this epidemiological investigation, and censoring occurred based on instances of CVD mortality. The 2006 and 2008 waves were combined to retrieve full data on target variables, as survey items corresponding to psychosocial stressors were administered to roughly half of the study sample in 2006 and 2008, respectively. The HRS had a total sample size of 43,399 participants in 2006–2008, and 7265 were working. Mortality data for the HRS cohort are available through 2018, representing a maximum follow-up period of 12 years. The size of the final analytic sample was 4046, and the process of sample size selection for the HRS cohort is shown in Figure 1.

### 2.2. Exposure Measures

The selected exposure measures serving as the independent variables in our investigation were designed to achieve a comprehensive investigation of work and nonwork related psychosocial stressors across the life course. In addition to sociodemographic characteristics and health behaviors such as age, sex, race, educational attainment, household income, smoking, alcohol consumption, and physical exercise at baseline, the key measures are detailed below.

### 2.3. ACEs

In the HRS study, ACEs were assessed via a series of detailed questions about the participant’s childhood, relationships with parents, and socioeconomic status (SES) in early life. The core components of ACEs identified in the scientific literature have been found to comprise a factorial structure with three key subdomains: (i) parental abuse, (ii) financial stress, and (iii) household dysfunction; this compartmentalization identifies the fundamental aspects of ACEs and offers a basic methodological framework for their analysis [43,44]. In the HRS cohort, ACEs were assessed in the Leave-Behind Questionnaire (LBQ), which was administered in two stages across the 2006 and 2008 cohorts, and has been used extensively for analyses of stress, childhood adversity, and health [45]. Physical abuse from parents, household dysfunction, and financial stress were assessed with a variety of items (see Table A1). A binary variable was constructed based on high or low ACEs exposure, and participants with two or more ACEs were categorized as experiencing high exposure to ACEs. 

### 2.4. Social Isolation

While the construct of social isolation is conceptually defined as the condition of being alone, the objective, direct indicators of social isolation are the frequency of social contacts and living status (i.e., living alone) [46,47,48]. It has been well established that the Berkman-Syme Social Network Index can be used to measure social isolation, asking if participants had regular contact with (i) both family members and friends (meeting in person, speaking on the phone, or writing/emailing), (ii) if they attended religious services, (iii) if they belonged to a social organization, club, or group, and (iv) whether they were married or lived alone [49]. In the HRS study, social isolation measures included assessments of marital status, contact with family and friends, religious service attendance, and meetings with programs, groups, and social clubs (see Table A1). Participants were assigned a score on a scale of zero to four depending on if they were married, if they had at least weekly contact with both family and friends, if they attended religious services at least monthly, and if they had at least monthly contact with a program, group, or social club. A social isolation score was obtained by reversing the resulting sum score, and a dichotomous variable for high and low social isolation was created; participants with a social isolation score of three or more were considered socially isolated. 

### 2.5. Job Strain 

Job strain is defined as per Karasek’s Job Demand-Control model, namely the combination of high job demands with low job control [50]. In general, the Job Content Questionnaire (JCQ) is used to measure job strain [51]. Although the original survey items within the JCQ were not applied in the HRS study, psychosocial work characteristics were measured in detail, encompassing job demands and job control (see Table A1). Job demands and control were dichotomized into high and low groups by their median scores (9 for both job demand and job control), and a binary job strain variable was constructed based on high job demands and low job control. 

### 2.6. Outcome: Cardiovascular Disease Mortality

The dependent variable and target outcome of interest was CVD mortality during follow-up, including deaths from heart disease and stroke. Furthermore, data on self-reported physician-diagnosed CVD at baseline in either 2006 or 2008 were used to exclude participants with pre-existing CVD (by affirmative response to any of the questions “In the last two years, have you had a heart attack or myocardial infarction?”, “Has a doctor ever told you that you had a stroke, heart attack, coronary heart disease, angina, congestive heart failure, or other heart problems?”), thus increasing the robustness of CVD mortality analyses. In the HRS study, CVD mortality data through 2018 were available through linkages to the National Death Index, and additionally through biennial exit interviews with surviving household members, with CVD mortality cases defined by ICD-10 codes I121-129, together with timing of death [52]. 

### 2.7. Statistical Analysis

The investigation of psychosocial work and nonwork-related stressors and their associations with risk of CVD mortality is the central outcome for this research project. First, descriptive statistics were generated, and relative frequencies were examined for characteristics of the sample populations at baseline. Second, the prospective associations of ACEs, social isolation, and job strain at baseline were assessed via Cox proportional hazards regression models, first separately and without mutual adjustment between all psychosocial exposures, and then together, with mutual adjustment, in order to assess potential independent effects between exposures. The results were expressed as hazard ratios (HRs) and 95% confidence intervals (CIs). Multivariable models were calculated in three steps: Model I adjusted for age and sex; Model II included further adjustment for race, educational attainment, and annual household income; Model III additionally adjusted for current smoking, alcohol consumption, and physical exercise. Hypothesis tests were two-sided at the 5% α level. Furthermore, we tested for interactions among ACEs, social isolation and job strain with CVD mortality as the outcome. We also examined associations of ACEs, social isolation, and job strain at baseline with all-cause mortality, to increase the number of endpoint events and assess whether the effect of these stressors on CVD and non-CVD mortality exhibited a similar direction or magnitude (see Table A2 and Table A3).

## 3. Results

### 3.1. Characteristics of the Sample Populations

The characteristics of the sample population in the HRS study are displayed in Table 1. In the 2006 and 2008 waves of the HRS study, the baseline time-point used for statistical analyses, the sample of 4046 working participants had roughly equal numbers of men and women, with a slightly greater proportion of women (53.34%), and were at a mean age of 61. They were mostly White (80.55%), with some Black participants (12.14%) and small numbers of Other races (7.32%). They had more limited levels of educational attainment, with 61.05% having completed high school or less, and generally had an annual household income of over $45,000. There was a low proportion of smokers (14.29%) and most participants had low or moderate alcohol consumption (96.59%) and high levels of physical activity (56.95%). The prevalence of high ACEs was 4.70%, while the prevalence of high social isolation was 7.19%, and the prevalence of high job strain was 16.73%. There were 59 cases of CVD deaths in the HRS sample. 

### 3.2. Associations of ACEs, Social Isolation, and Job Strain with Cardiovascular Disease Mortality

Table 2 shows the number of exposed participants, CVD mortality cases, and crude CVD mortality rates for each exposure group, and Table 3 and Table 4 demonstrate the results of the Cox proportional hazards regression models assessing the prospective associations of ACEs, social isolation, and job strain with CVD mortality, with Table 3 showing results without mutual adjustment, and Table 4 showing findings with mutual adjustment for psychosocial exposures.

In the HRS study, during 42,149 person-years of follow-up time across approximately 12 years of follow-up, 59 cases of CVD mortality were reported, constituting an aggregate CVD mortality rate of 1.40 per 1000 person-years. CVD mortality rates were 1.29 and 3.68 per 1000 person-years for participants with low and high levels of ACEs exposure, respectively. CVD mortality rates for social isolation were 1.32 and 2.59 per 1000 person-years for low and high levels of social isolation, respectively. For job strain, CVD mortality rates were 1.26 and 2.10 per 1000 person-years for participants with low and high levels of job strain exposure, respectively. The results of the interaction analyses of ACEs, social isolation and job strain with CVD mortality as the outcome did not indicate significance (Table 2).

In the regression models without mutual adjustment between exposures, compared to low ACEs exposure, high ACEs exposure was significantly associated with increased CVD mortality (fully-adjusted HR and 95% CI = 4.07 (1.79, 9.24)). While high social isolation was not significantly associated with CVD mortality, participants exposed to high social isolation exhibited elevated hazard estimates (fully-adjusted HR and 95% CI = 1.87 (0.84, 4.15)). High levels of job strain were also significantly associated with CVD mortality, compared to low levels of job strain (fully-adjusted HR and 95% CI = 2.27 (1.24, 4.15)) (Table 3). 

Mutually adjusted regression models revealed slightly attenuated associations of these three psychosocial factors with increased CVD mortality, i.e., fully-adjusted HRs and 95% CIs were 3.67 (1.59, 8.48) for high ACEs, 1.62 (0.72, 3.66) for high social isolation, and 2.24 (1.21, 4.11) for high job strain, respectively (Table 4).

Analyses of all-cause mortality indicated a similar direction of associations, although with attenuated effect sizes (see Table A2 and Table A3).

## 4. Discussion

Using the data from the large, nationally representative, population-based HRS study, we examined the prospective associations of ACEs, social isolation, and job strain with CVD mortality. ACEs and job strain were significantly associated with CVD mortality in older employees, while social isolation was related to increased risk of CVD mortality but did not reach statistical significance. Mutual adjustment between the psychosocial exposures of ACEs, social isolation, and job strain did not indicate marked differences compared to models foregoing mutual adjustment. Together, these results suggest a notable influence of psychosocial stressors on deaths due to CVD. Our hypotheses were therefore supported by the findings.

The significant associations of ACEs with CVD mortality detected in the HRS sample are consistent with the literature on the adverse cardiometabolic health impacts of ACEs exposure. Review studies assaying evidence on ACEs and CVD later in life have reported that adults with high exposure to ACEs have a “more than 2-fold higher risk of developing CVD and an almost 2-fold higher risk of premature mortality” [21]. ACEs are widely implicated in cardiometabolic health conditions, including clinically proven risk factors for CVD such as hypertension [20,53,54]. The AHA has also published scientific statements acknowledging ACEs as a social determinant of CVD risk and cardiometabolic health outcomes and calling for targeted longitudinal investigations—this project suitably addressed this call for further research by providing survival analysis data from prospective cohort studies [20,55]. Such findings provide preliminary evidence in support of the “chains of risk” model, where an early stress exposure might lead to further stressful experiences later in life [56,57,58]; ACEs may lead to more adverse working conditions and augment perceptions of stressful work environments in adulthood [59]. ACEs may also lead to impaired social functioning in adolescence and adulthood, based on neurobiological evidence showing disruptions of neural networks involved in social functioning [60]. This has been documented to lead to deficits in processing social cues and body language such as facial expressions, with neuroimaging studies of physically abused children showing effects of bias towards angry faces and negative emotions [61,62,63,64]. Alternative explanations explored by life-course exposure models [56,65] and the “biological embedding” model of stress and disease posit that early childhood is a critical period that greatly influences responses to environmental and psychosocial stressors later in life [66,67]. Long-term biological consequences of ACEs include disruption of regulatory and homeostatic mechanisms, including the immune, metabolic, neuroendocrine, and autonomic nervous systems [68]. ACEs have been shown to produce chronic inflammation, as evidenced by elevated proinflammatory signaling molecules such as interleukin-6 (IL-6), C-reactive protein (CRP), and fibrinogen [69,70,71]. Notably, recent evidence showing efficacy of CVD medications such as PCSK9 inhibitors in ameliorating CVD biomarkers such as low-density lipoprotein cholesterol (LDLC) also demonstrated reductions in psychological distress and improved quality of life, indicating a potential pathway for interventions targeting psychosocial stress and CVD [72]. 

While associations of social isolation with CVD mortality did not reach statistical significance in the current analyses, elevated hazard estimates were observed amongst participants with high social isolation, which is in accordance with prior investigations of such psychosocial exposures and CVD. One potential explanation for the insignificant findings may be the fact that the participants were employees, as opposed to retirees or the general population—employed participants may experience social contacts through the work environment, based on engagement in work tasks and with colleagues. Therefore, workplace social support and task involvements may offset the negative effects of nonwork-related social isolation. Social isolation has a plethora of effects that undermine an individual’s health and overall well-being, with recent data aggregated in a comprehensive report disseminated by the U.S. Surgeon General [73]. Indeed, evidence illustrates a strong connection between social isolation and heart disease and stroke outcomes, with many studies demonstrating significant increases in morbidity and mortality [74,75,76,77]. Estimates for the effect of social isolation range from 30–50% for increased risk of heart disease and stroke [16,22], as well as drastically increased risk of hospitalization (68%) and emergency department visits (57%) related to chronic cardiovascular conditions [78]. The increased risk of death and healthcare use associated with social isolation clearly demonstrate clinical relevance in CVD. There is also a pragmatic element of social isolation that is relevant to CVD outcomes—instances of fatality from myocardial infarction are less likely when household members or social contacts are able to provide immediate help and calls for medical attention. In context of population-level demographic trends apparent in the U.S., where the percentage of single-person households reached 29% in 2022, these data call attention to social isolation as an urgent and pressing issue of public health significance. The rapidly increasing prevalence of social isolation in the U.S. and the burgeoning body of literature on associated cardiovascular health consequences prompted the AHA to issue a 2022 statement underscoring social isolation as a “common, yet underrecognized determinant of cardiovascular health and brain health” [77]. 

Job strain has also been strongly substantiated as a major risk factor for CVD and worsened CVD mortality outcomes. A review of evidence from over 600,000 adults from 27 studies across the U.S., Europe, and Japan found that occupational stressors such as job strain were associated with an up to 40% increased risk of incident coronary heart disease and stroke [25]. Meta-analyses of over 190,000 participants from 13 European cohort studies found robust associations of job strain with coronary heart disease and stroke, with the population attributable risk for coronary heart disease due to job strain estimated at 3.4% [79,80]. Similarly, a multicohort study drawing on data from Finland, France, Sweden, and the United Kingdom with over 100,000 individuals found evidence for excess CVD mortality risk among participants with prevalent cardiometabolic diseases at baseline [81]. These data highlight the importance of including novel and psychosocial exposures such as social isolation and job strain in investigations of CVD mortality in working populations. 

### 4.1. Strengths

The major strengths of this study are founded upon the large, population-based, nationally representative sample population among older employees. The sample included strong representation of sociodemographic ranges and occupations and provided a long follow-up length of 12 years, thus addressing limitations of previous studies and the lack of longitudinal cohort data highlighted by the AHA’s 2022 scientific statement on ACEs [20]. Additionally, the older mean age of the sample (61) and long follow-up period likely allowed for the manifestation of CVD deaths and thus detection by the statistical analyses, as the risk of cardiometabolic conditions increases with age [82,83]. In a study of over 130,000 Chinese adults, socioeconomic and psychosocial risk factors were found to be prominent determinants of CVD outcomes among participants aged 55 years or above, accounting for 27% of the population attributable risk for mortality [84].

Furthermore, the measures used were detailed and reliable; job strain measures generally adhered to Karasek’s well-established model job strain [50], and the well-validated Berkman-Syme Social Network Index was utilized to assess social isolation [85]—one study using data from the HRS cohort utilizing this approach, showed promising findings regarding outcomes including all-cause mortality and excess Medicare spending [86]. Job strain data from the HRS study have been successfully implemented in studies of hypertension, depression, drinking behavior, memory function, and long-term mortality [87,88,89,90]. ACE measures across the studies were also rich and detailed, adequately encompassing the three-factor structure of ACEs with assessments of abuse, household dysfunction, and financial stress. The HRS dataset also provided data on sociodemographic characteristics and health-related behaviors that were used to account for potential confounders.

### 4.2. Limitations

There are several limitations of this study that may have impacted the findings. While the reliability of adult retrospective reports of ACEs has also come under question due to potential recall bias, studies assessing retrospective reports have observed robust test-retest reliability ranging from 0.45 to 0.90, and adequate kappa coefficients ranging from 0.52 to 0.72 [91,92]. Similarly, because all exposure data was collected at baseline, the results may be influenced by exposure misclassification bias due to the possibility of changes in social isolation and job strain during follow-up. In terms of data quality, it is also possible that the use of surveys and administrative data in assessments of CVD and CVD mortality introduced potential bias in the analyses. Furthermore, a substantial number of participants were excluded due to missing data on psychosocial factors, given the special research design in the HRS study, i.e., roughly half of the study sample in 2006 and 2008 were invited to participate in psychosocial surveys, respectively, and we pooled them together to form our baseline sample. As a result, we were unable to rule out the possibility of non-invitation or non-response. Hence, caution should be exercised when generalizing our findings to all older U.S. employees. Additionally, while the sample population was large and included a broad range of demographic strata, most participants were White, with limited representation of Black and other racial groups, limiting the generalizability of the results. Future studies with better representation of racial and ethnic minorities would allow for further extrapolation and comparison of results with broader populations. Finally, while these findings showed significant associations of ACEs and job strain with CVD mortality, they raise further questions regarding the biological mechanisms underlying such observations and the means by which such exposures impact cardiometabolic health. Hence, the contributions of ACEs and job strain to manifested diseases and relevant biomarkers deserve further investigation.

## 5. Conclusions

With the large, nationally-representative, population-based HRS cohort among older employees, the psychosocial exposures of ACEs and job strain were significantly associated with increased risk of CVD mortality. These findings add to the weight of evidence identifying psychosocial exposures as clinically relevant risk factors for CVD.

## Figures and Tables

**Figure 1 medicina-59-01304-f001:**
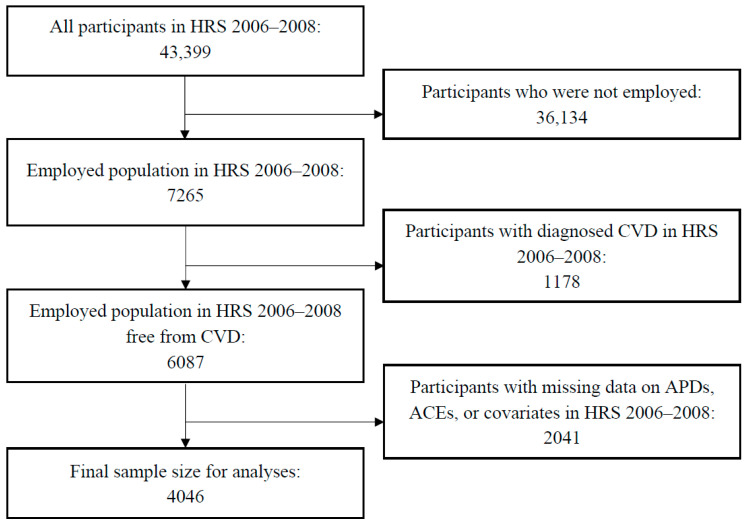
Sample Size Selection Flowchart for the HRS Study (2006–2008).

**Table 1 medicina-59-01304-t001:** Characteristics of the Sample Population in the Health and Retirement Study (HRS), 2006–2008 (*N* = 4046).

Variables (*N*, %)	
Age (mean, SD)	61.16 (6.99)
Sex	
Male	1888 (46.66)
Female	2158 (53.34)
Race	
White	3259 (80.55)
Black	491 (12.14)
Other	296 (7.32)
Educational attainment	
University or more	1229 (30.38)
Some college	347 (8.68)
High school or less	2470 (61.05)
Household income (annual U.S. dollars)	
<45,000	1594 (39.40)
45,000–89,999	1358 (33.56)
≥90,000	1094 (27.40)
Smoking status	
No	3468 (85.71)
Yes	578 (14.29)
Alcohol consumption	
Low or moderate drinking	3908 (96.59)
Heavy drinking	138 (3.41)
Physical activity	
High	2304 (56.95)
Moderate	481 (11.89)
Low	1261 (31.71)
Adverse childhood experiences	
Low	3856 (95.30)
High	190 (4.70)
Social isolation	
Low	3755 (92.81)
High	291 (7.19)
Job strain	
Low	3369 (83.27)
High	677 (16.73)
Cardiovascular disease deaths	
No	3987 (98.54)
Yes	59 (1.46)

**Table 2 medicina-59-01304-t002:** Number of Exposed Participants, CVD Mortality Cases, and CVD Mortality Rates by Exposure Group in the Health and Retirement Study (*N* = 4046).

	Number of Exposed Participants (Number of Cardiovascular Mortality Cases)	Cardiovascular Disease Mortality Rate (Per 1000 Person-Years)
Adverse Childhood Experiences		
Low	3856 (52)	1.29
High	190 (7)	3.68
Social isolation		
Low	3755 (52)	1.32
High	291 (7)	2.59
Job strain		
Low	3369 (44)	1.26
High	677 (15)	2.10

**Table 3 medicina-59-01304-t003:** Separate Prospective Associations of Adverse Childhood Experiences (ACEs), Social Isolation, and Job Strain with Cardiovascular Disease Mortality in the Health and Retirement Study (HRs and 95% CIs) (*N* = 4046).

	Model I	Model II	Model III
Adverse Childhood Experiences			
Low	1.00	1.00	1.00
High	4.17 (1.87, 9.26) **	4.46 (2.00, 9.98) **	4.07 (1.79, 9.24) **
Social isolation			
Low	1.00	1.00	1.00
High	1.94 (0.88, 4.27)	1.96 (0.89, 4.32)	1.87 (0.84, 4.15)
Job strain			
Low	1.00	1.00	1.00
High	2.52 (1.39, 4.59) **	2.54 (1.39, 4.61) **	2.27 (1.24, 4.15) **

CI, confidence interval; HR, hazard ratio. Cox proportional hazards regression, ** *p* < 0.01. Adverse childhood experiences, social isolation, and job strain were not mutually adjusted for each other. Model I: adjustment for age and sex at baseline. Model II: Model I + additional adjustment for race, educational attainment, and household income at baseline. Model III: Model II + additional adjustment for smoking, alcohol consumption, and physical exercise at baseline.

**Table 4 medicina-59-01304-t004:** Independent Prospective Associations of Adverse Childhood Experiences (ACEs), Social Isolation, and Job Strain with Cardiovascular Disease Mortality in the Health and Retirement Study (HRs and 95% CIs) (*N* = 4046).

	Model I	Model II	Model III
Adverse Childhood Experiences			
Low	1.00	1.00	1.00
High	3.65 (1.61, 8.26) **	3.83 (1.67, 8.79) **	3.67 (1.59, 8.48) **
Social isolation			
Low	1.00	1.00	1.00
High	1.66 (0.74, 3.74)	1.68 (0.75, 3.80)	1.62 (0.72, 3.66)
Job strain			
Low	1.00	1.00	1.00
High	2.48 (1.36, 4.53) **	2.46 (1.34, 4.51) **	2.24 (1.21, 4.11) **

CI, confidence interval; HR, hazard ratio. Cox proportional hazards regression, ** *p* < 0.01. Adverse childhood experiences, social isolation, and job strain were mutually adjusted for each other. Model I: adjustment for age and sex at baseline. Model II: Model I + additional adjustment for race, educational attainment, and household income at baseline. Model III: Model II + additional adjustment for smoking, alcohol consumption, and physical exercise at baseline.

## Data Availability

Data used in this study are publicly available at https://hrsdata.isr.umich.edu/data-products/rand (accessed on 6 January 2023).

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
