# Peer review of "Adverse Childhood Experiences, Social Isolation, Job Strain, and Cardiovascular Disease Mortality in U.S. Older Employees"

_medicina, 2023, doi:10.3390/medicina59071304_

Round 1

Reviewer 1 Report

I have had the opportunity to review the article entitled "Adverse Childhood Experiences, Social Isolation, Job Strain, and Cardiovascular Disease Mortality in U.S. Older Employees". The manuscript is original and of potential interest. I have the following comments:

1) The paper is acceptable, but the presentation of the manuscript should be improved. Please correct typos and grammatical errors in the manuscript.

2) The "Introduction" section is too long and somewhat redundant. I recommend that the length of this section be significantly reduced to improve readability. 

3) The use of surveys and administrative data for the assessment of CVD and CVD mortality may introduce potential bias in the analysis. The authors should acknowledge this aspect among the limitations.

4) In addition to CVD mortality, the authors should assess the impact of ACEs, social isolation, and job strain on all-cause mortality. Separate assessment of non-CVD mortality would also be of interest to understand whether or not the effect of these stressors on CVD and non-CVD mortality has a similar direction or magnitude. The analysis of all-cause mortality would also be useful to increase the number of endpoint events, which is limited for CVD mortality (59 deaths overall). 

5) The use of sub-headings in the "Methods" section would be useful to improve readability, for example: Definition of ACEs, Study Endpoints, Statistical Analysis, etc. 

6) Novel therapies have been shown to reduce psychological stressors and improve quality of life in patients at high risk for CVD. I recommend discussing this concept making reference to PMID: 30913901, which provides recent evidence on this topic. 

The presentation of the manuscript should be improved. 

Author Response

Please see our response in the enclosed file. Thanks. 

Author Response

(The authors gave the same response as above.)

Reviewer 3 Report

The manuscript by Matthews and Li is a longitudinal population-based study on the impact of stress on cardiovascular disease. Authors analysed three questionnaires on social isolation, Adverse Childhood Experiences, and work strain, finding an association between the last two and the development of CVD. 

The manuscript is overall well written and the content is of merit.

I suggest Authors to significantly shorten the introduction section for the sake of readability and clearness. 

Author Response

(The authors gave the same response as above.)

Round 2

Reviewer 1 Report

I have no further comments.

Reviewer 2 Report

The authors reviewed and resolved all the comments